# Occurrence of *Rickettsia* spp., *Hantaviridae*, *Bartonella* spp. and *Leptospira* spp. in European Moles (*Talpa europaea*) from the Netherlands

**DOI:** 10.3390/microorganisms11010041

**Published:** 2022-12-22

**Authors:** Tryntsje Cuperus, Ankje de Vries, Ryanne I. Jaarsma, Hein Sprong, Miriam Maas

**Affiliations:** Centre for Infectious Disease Control, National Institute for Public Health and the Environment (RIVM), Antonie van Leeuwenhoeklaan 9 Bilthoven, 3720 BA Utrecht, The Netherlands

**Keywords:** *Talpa europaea*, mole, zoonotic pathogens, zoonoses, epidemiology

## Abstract

The European mole (*Talpa europaea*) has a widespread distribution throughout Europe. However, little is known about the presence of zoonotic pathogens in European moles. We therefore tested 180 moles from the middle and the south of the Netherlands by (q)PCR for the presence of multiple (tick-borne) zoonotic pathogens. Spotted fever *Rickettsia* was found in one (0.6%), *Leptospira* spp. in three (1.7%), *Bartonella* spp. in 69 (38.3%) and *Hantaviridae* in 89 (49.4%) of the 180 moles. Infections with *Anaplasma phagocytophilum*, *Babesia* spp., *Neoehrlichia mikurensis*, *Borrelia* spp., *Spiroplasma* spp. and *Francisella tularensis* were not found. In addition, in a subset of 35 moles no antibodies against *Tick-borne encephalitis virus* were found. The obtained sequences of *Bartonella* spp. were closely related to *Bartonella* spp. sequences from moles in Spain and Hungary. The *Hantaviridae* were identified as the mole-borne Nova virus, with high sequence similarity to sequences from other European countries, and Bruges virus. Though the zoonotic risk from moles appears limited, our results indicate that these animals do play a role in multiple host-pathogen cycles.

## 1. Introduction

The European mole (*Talpa europaea*) is a small insectivorous mammal with a fossorial lifestyle. Moles belong to the order of *Eulipotyphla* (insectivores) together with hedgehogs, shrew-moles and true shrews. The distribution of the European mole runs from Britain and Northern Spain eastwards deep into Russia [1]. In Western Europe, including the Netherlands, moles are common and inhabit a wide range of habitats, including urban areas. Although not frequent, humans come into contact with moles. This contact creates a potential health risk, because insectivores are known to carry a diversity of zoonotic pathogens, e.g., *Leptospira* spp. and *Bartonella* spp. [2,3,4]. However, little is known about the presence of zoonotic pathogens in European moles. As far as the authors know, there are no literature reports describing human infectious disease cases where moles were considered the source. It is unclear whether this is caused by the absence of zoonotic pathogens in moles, the low contact rates with humans or the mild, non-specific symptoms caused by mole-borne zoonoses.

Recent studies reported the presence of a potentially novel *Bartonella* species [3,4,5], *Babesia venatorum* [6], a spotted fever *Rickettsia* [3] and *Toxoplasma gondii* [7] in moles from Spain, Slovakia, Germany, Hungary and the Netherlands respectively. *Borrelia burgdorferi* sensu lato, *Anaplasma phagocytophilum, Leptospira* spp., *Coxiella burnetii* and *Francisella tularensis* were not found [8,9,10]. However, the number of moles tested in these studies was in most cases relatively small, making statements about the role of the mole in the transmission cycles for these pathogens difficult.

Orthohantaviruses (family: *Hantaviridae*, order: *Bunyavirales*) are more extensively studied in moles. Multiple orthohantaviruses can cause disease in humans, most notably hantavirus hemorrhagic fever with renal syndrome (HFRS) in Eurasia and hantavirus pulmonary syndrome (HPS) in the Americas [11]. In addition to rodents, in recent decades numerous orthohantaviruses were found in hosts from the orders *Eulipotyphla* (insectivores) and *Chiroptera* (bats) [12]. Bruges virus (BRGV, species: *Bruges orthohantavirus*), was described in moles from Belgium, Germany and the UK [13]. Furthermore, Nova virus (NVAV, species: *Nova mobatvirus;* family: *Hantaviridae*)*,* was described in a mole from Hungary [14] and in moles from France, Poland and Belgium [15,16,17]. It is as yet unclear whether BRGV or NVAV can cause disease in humans.

The aim of the current study was to assess the occurrence of a range of potentially zoonotic pathogens in a large sample of moles from the Netherlands. We therefore tested 180 moles from different locations for *Hantaviridae*, *Leptospira* spp., *Bartonella* spp., *Rickettsia* spp., *Anaplasma phagocytophilum*, *Babesia* spp., *Borrelia* spp., *Neoehrlichia mikurensis, Spiroplasma* spp. and *Francisella tularensis*. In addition, a subset of moles was tested for antibodies against *Tick-borne encephalitis virus* (TBEV).

## 2. Material and Methods

### 2.1. Sample Collection

Across different locations in the Netherlands, 180 moles that had been lethally trapped by professional molecatchers, were collected (Figure 1). Trapping had been conducted as part of pest control, thus no additional permits or ethical permission were required for the study. Moles in the central part of the Netherlands (location Urk) were captured in the spring of 2018. Moles from the southern region (locations Lage Zwaluwe and Bergen op Zoom) and South-Eastern region (locations Gennep and Schimmert) were captured in the spring of 2019. Between Urk and the southern capture locations there is a distance of 135–230 km. Between the different southern capture locations there is a distance of maximum 135 km. The information received from the molecatchers regarding the habitat ranged from detailed GPS coordinates to general description of the capture location. After capture, moles were stored at −20 °C until dissection. Tissues (lung, kidney, liver and spleen) were collected and stored at −80 °C. Heart fluid was collected as described previously [18] and stored at −20 °C. Body weight and sex were recorded (Appendix A).

### 2.2. Nucleic Acid Extraction, qPCR and Sequencing

Total nucleic acid was isolated from lungs, kidney and spleen from each animal using the MagNAPure Total NA isolation kit (Roche Diagnostics GmbH, Mannheim, Germany) as described previously [20]. Previously established qPCR-based methodologies were used for the detection of genetic material from the following pathogens: *Borrelia burgdorferi* s.l. [21], *Borrelia miyamotoi* [22], *Neoehrlichia mikurensis* [23], *Anaplasma phagocytophilum* [24], *Babesia microti* [25], *Babesia* spp. from clade X, which has been designed to detect *B. divergens*, *B. venatorum*, *B. capreoli* and *B. odocoilei* [26], spotted fever group *Rickettsia* [27], *Bartonella* spp. [28], *Francisella tularensis* [29], *Spiroplasma* spp. [30] and *Leptospira* spp. [31]. Primer sequences can be found in Appendix A.

All qPCRs were carried out on a LightCycler 480 (Roche Diagnostics Nederland B.V., Almere, the Netherlands) in a final volume of 20 μL with iQ Multiplex Powermix (Bio-Rad Laboratories, Veenendaal, the Netherlands), 3 μL of sample DNA, 0.2 μM for all primers and different concentrations for probes. Positive controls and negative water controls were used on every plate tested. Analysis was performed using the second derivative calculations for Cp (crossing point) values. For overflow of fluorescence from dyes that were used, a colour compensation was conducted. Curves were assessed visually.

For hantaviruses, an RT-PCR was performed to detect the conserved L-segment, followed by a nested PCR as described [32]. Samples positive for *Bartonella* spp. were subjected to a conventional PCR of a fragment from the citrate synthase gene (gltA; [33]). PCR products were purified for sequencing with ExoSAP-IT PCR clean-up (Isogen Life Science, Utrecht, The Netherlands), followed by Sanger sequencing (Baseclear, Leiden, The Netherlands). Obtained sequences were submitted to Genbank with accession numbers OM513926-OM513933 (*Bartonella* spp.) and OM513918-OM513925 (orthohantaviruses). Samples positive for *Leptospira* spp. were subjected to a melt-curve analysis according to Ahmed et al. [31] to determine the *Leptospira* species.

### 2.3. TBEV Serology

In a subset of moles from Urk (n = 35), heart fluid was analyzed for antibodies against TBEV using the EIA TBE Virus IgG kit (TestLine Clinical Diagnostics, Brno, Czech Republic) according to the manufacturer’s instructions. As a conjugate Protein G IgG HRP (Thermo Fisher Scientific, Landsmeer, The Netherlands) was used.

### 2.4. Phylogenetic Analysis

Multiple sequence alignments for the hantavirus L-segment and *Bartonella* spp. gltA sequences were obtained with the MAFFT algorithm [34]. Maximum likelihood phylogenetic trees were generated by IQtree [35] with 10,000 ultrafast bootstrap replicates [36]. Final trees were visualized in the FigTree v.1.4.4. program [37].

## 3. Results

### 3.1. Apparent Pathogen Prevalence

In three moles (1.7%, 95% CI 0.4–4.8) *Leptospira* spp. DNA was detected in kidney samples (Figure 1, Table 1 and Appendix A). All *Leptospira*-positive moles originated from the capture location Urk. Two moles were found to be infected with *Leptospira interrogans*, while *Leptospira kirschneri* was found in one mole.

*Bartonella* spp. DNA was detected in all locations, with an overall prevalence of 38.3%, 95% CI 31.2–45.9 (Figure 1, Table 1 and Appendix A). *Bartonella* spp. prevalence ranged from 15% to 80% at the different capture locations. No difference in *Bartonella* spp. prevalence was seen between male (39.7%) and female moles (37.6%).

Almost half of the tested moles (49.4%, 95% CI 41.9–57.0), originating from all locations tested, were found to be infected with hantavirus (Figure 1, Table 1 and Appendix A). Hantavirus prevalence for moles captured in 2018 around Urk (50.4%) did not differ significantly from the prevalence of those captured in 2019 from southern regions (47.3%). Neither was a difference seen in prevalence between male (47.4%) and female (50.5%) moles.

Spotted fever *Rickettsia* DNA was found in one mole (0.6%, 95% CI 0.01–3.1) from the Lage Zwaluwe location. This animal was also infected with both hantavirus and *Bartonella* spp. In total, in 40 moles a mixed infection of hantavirus and *Bartonella* spp. was found (22.2%, 95% CI 16.4–29.0). In one of these moles a triple infection of *Leptospira* interrogans, hantavirus and *Bartonella* spp. was detected (Appendix A).

All moles were negative for *Anaplasma phagocytophilum*, *Babesia* spp., *Neoehrlichia mikurensis*, *Borrelia* spp., *Spiroplasma* spp. and *Francisella tularensis* DNA. The selection of moles (n = 35, from Urk) tested for antibodies against TBEV were all negative.

### 3.2. Phylogenetic Analysis

A phylogenetic tree made from the sequenced fragment of the gltA gene of *Bartonella* showed that our sequences clustered in two groups with two nucleotides difference (Figure 2). Sequences from both clusters were found at all capture locations. The highest similarity of our sequences was found with uncultured *Bartonella* from moles in Spain [5] (100% similarity) and uncultured *Bartonella* from multiple animals (99.7%, mole, hedgehog, weasel and house mouse) from Hungary [3].

The phylogenetic tree of the hantavirus L-segment shows sequences falling into multiple clusters (Figure 3). Two separate clusters of sequences are seen from moles captured around Urk. The similarity between sequences from one capture location was 83.1–100%.

All but one of the sequences were identified as NVAV. One mole from the Gennep capture location was found to be positive for BRGV. Between all the Dutch NVAV sequences, the lowest similarity was 86.2%. Similarity with NVAV sequences from other European countries was 81.1–88.7%. The BRGV sequence clustered with BRGV sequences from Belgium and Germany and showed 82.6% and 83.1% similarity, respectively.

## 4. Discussion

In a population of 180 moles, *Leptospira* spp., *Bartonella* spp., *Hantaviridae* and spotted fever *Rickettsia* were detected, suggesting a potential role for moles in zoonotic pathogen transmission to humans.

*Leptospira* spp. was found in three moles, which to our knowledge is the first description of this pathogenic bacterium in moles. Previously, moles from Germany were tested for *Leptospira*, but none were found positive [9]. However, *Leptospira* has been described in related insectivore hosts, such as shrews [2,38,39]. Two Leptospira species were found: *Leptospira interrogans* and *Leptospira kirschneri*. Both species are pathogenic to humans, with *Leptospira interrogans* being the main cause of human leptospirosis and *Leptospira kirschneri* being the causative agent of recent outbreaks in strawberry pickers in Germany [40,41]. The Netherlands has an average yearly incidence of human leptospirosis of 0.25 cases/100,000 population, and incidence is highest in the northern regions of the country [42]. This agrees with our finding of *Leptospira* spp. in the moles tested from Urk, the most northern region in this study.

Multiple species of *Bartonella* are causative agents of human diseases including Carrion’s disease and cat-scratch disease [43]. In the last two decades, association of human disease with different *Bartonella* species has continued to grow. *Bartonella* spp. was found in 38.3% of Dutch moles. This bacterium was previously reported in 15 out of 21 moles from Spain [5].In addition, *Bartonella* was also found in small numbers of moles from Slovakia and Hungary [3,4]. Sequences from the current study were closely related to the *Bartonella* sequences from Spain and Hungary. Possibly, these sequences could be part of a yet unidentified, mole-associated *Bartonella* species, that is found in at least three countries across Europe. The pathogenic potential for humans of the *Bartonella* spp. described here is unknown. However, the sequences found are genetically quite similar to *Bartonella* spp. that have been shown to infect humans [44,45]. In addition, it has been hypothesized that any species of *Bartonella* is capable of infecting humans [46]. For zoonotic infection to occur, frequent exposure to moles is essential, which is not expected to be the case for the general public.

Similar to reports from other European countries, NVAV was found in Dutch moles. Most of the Dutch NVAV sequences cluster away from NVAV sequences from other countries. Also, multiple distinct clusters of Dutch sequences from different geographical locations are seen, as reported previously for NVAV from France and Belgium [15,17]. The apparent prevalence of 49.4% found in our sample is similar to the prevalence in Belgium (53%) [17], but slightly lower compared to NVAV prevalences found in Poland (66%) [16] and France (65%) [15]. The high prevalence of NVAV infection in European moles suggests a well-established host-pathogen relationship and enzootic virus transmission. Interestingly, only one mole in our sample (0.6%) was found to be infected with BRGV. The prevalence of BRGV in Belgian moles was 4.6%, but no BRGV was found in moles from France or Poland [13], raising questions about the distribution and transmission efficiency of this virus.

At present, it is unclear whether NVAV or BRGV are pathogenic to humans. The lack of a small animal model able to mimic human hantavirus pathology makes it more difficult to answer this question. Experimental infection of mice with NVAV does lead to weight loss, hyperactivity and hind-limb paralysis [47]. In addition, some etiological evidence has been published linking non-rodent borne hantaviruses to human infections and possibly to human disease [48,49]. Currently, most hantavirus cases in the Netherlands are caused by Puumala virus, though differentiation between the various hantaviruses with the current diagnostics is not always performed [50].

The moles in our study were negative for a range of tick-borne pathogens, indicating that they could have no or only a minor role in the ecology of these pathogens. Moles from Spain and Hungary were previously tested for *Borrelia* spp., *Rickettsia* spp. and *Anaplasma phagocytophilum* and similar to our results, no infections with *Borrelia* or *Anaplasma* were found [3,8]. However, *Rickettsia* spp. was found in a single mole from Hungary [3], and also in our study a spotted fever group *Rickettsia* was found in one mole.

In moles from Slovakia, TBEV and TBEV antibodies were detected [51]. These moles were trapped in a well-known focus area of TBEV. In the Netherlands, TBEV has been detected in human patients, ticks and wildlife since 2016. A convenience sample of 35 moles from Urk was tested as part of another study on TBEV in the Netherlands. The capture location of Urk is close to a location where TBEV has been detected in ticks [52], however no antibodies against TBEV were found in the moles. Because of the limited presence of TBEV in the Netherlands and the low likelihood of finding TBEV in moles, no further moles were tested. In addition, very few tick-borne pathogens were found in the moles tested, similar to previous studies in European moles [8]. Possibly the European mole plays little to no role in tick-host-pathogen cycles. If so, this makes the European mole distinctive among land mammals.

This study addresses the circulation of potentially (tick-borne) zoonotic pathogens in moles. Because convenience samples from lethal trappings were used, no systematic approach could be followed to assess the true prevalence of the source population. Therefore, we were only able to assess the apparent prevalence of the samples of subjects analyzed. Also, a potential limitation of the study might be the use of a limited set of organs to test for the presence of pathogenic DNA and RNA. Kidney tissue was used to test for *Leptospira* spp., the lungs to test for hantaviruses and the spleen for all other pathogens. Testing more or other tissues could have led to the detection of different pathogens, for example testing of the skin for tick-borne pathogens. Storage of carcasses is ideally at −80 °C, which for practical reasons was not done in this study. This could have negatively affected pathogen detection. However, since hantaviruses were detected in all three batches of moles, this does not seem to have severely affected the carcasses. Furthermore, our investigation only detected the DNA or RNA of the described pathogens, but not their infectivity.

Unfortunately, no additional information was available about the habitats in which the moles had been captured, other than a general description of the capture locations. Therefore, no in depth analyses could be performed regarding environmental factors.

With this study we have taken a first step to elucidate the role of moles in host-pathogen cycles by testing for a range of potentially zoonotic pathogens. Notably, we report the detection of *Leptospira* in moles. In addition, high apparent prevalences of *Bartonella* and NVAV were found. Although for both of these pathogens the potential for human disease is as yet unclear, we recommend increasing awareness among physicians and public health workers about exposure to moles in individuals presenting with unusual clinical syndromes.

## Figures and Tables

**Figure 1 microorganisms-11-00041-f001:**
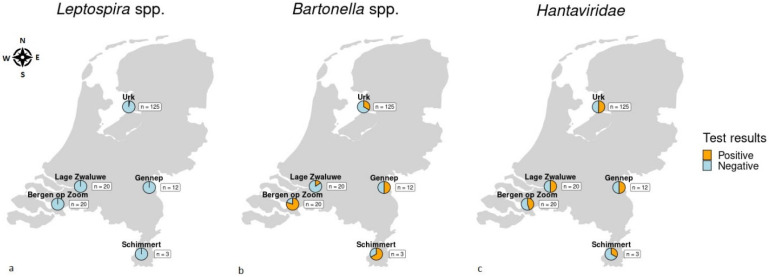
Maps of capture locations of moles (*Talpa europaea*) and proportion of (**a**) *Leptospira* spp., (**b**) *Bartonella* spp. and (**c**) *Hantaviridae* positive animals at each location. Maps were designed in R [19].

**Figure 2 microorganisms-11-00041-f002:**
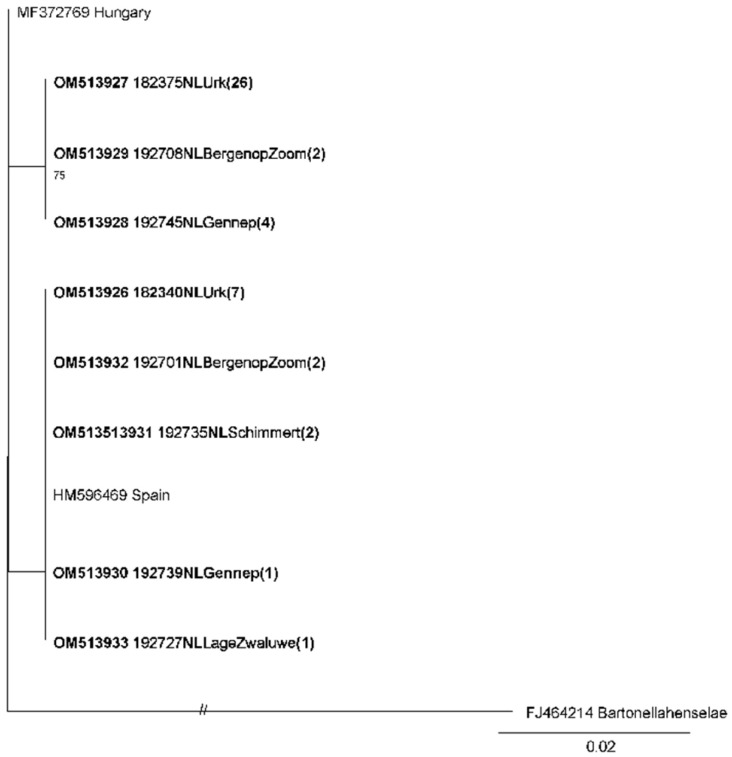
Maximum-likelihood phylogenetic tree for partial *Bartonella* gltA gene sequences (258 nt). Sequences from this study are shown in bold (Genbank accession numbers OM513926-OM513933). One representative sequence for each capture site was chosen for this tree, unless sequences from one capture site were from different branches. In parentheses the number of identical sequences in a specific branch is indicated. Numbers along branches are bootstrap values, only bootstrap support of >70% is shown. The scale bar indicates nucleotide substitutions per site.

**Figure 3 microorganisms-11-00041-f003:**
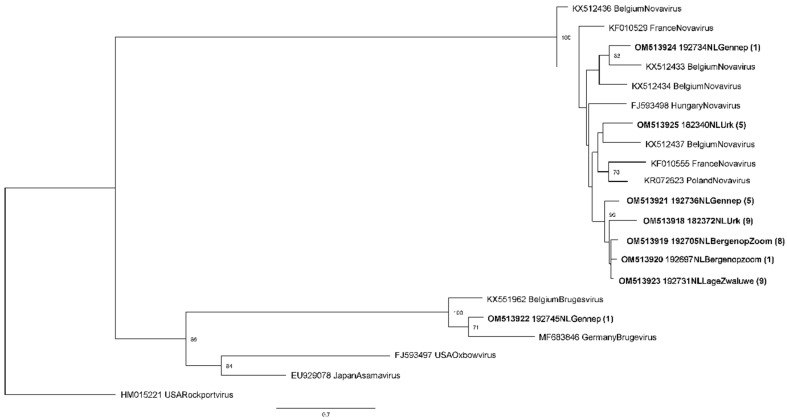
Maximum-likelihood phylogenetic tree for hantavirus partial L-segment sequences (353nt). Sequences from this study are shown in bold (Genbank accession numbers OM513918–OM513925). One representative sequence for each capture site was chosen for this tree, unless sequences from one capture site were from different branches. In parentheses the number of similar sequences in a specific branch is indicated. Numbers along branches are bootstrap values, only bootstrap support of >70% is shown. Scale bar indicates nucleotide substitutions per site.

**Table 1 microorganisms-11-00041-t001:** Apparent prevalence of tested pathogens in moles (*Talpa europaea*) from different capture locations.

	No. Positive Moles per Capture Location (Prevalence (%), 95% Confidence Interval)					
Pathogen *	Urk (n = 125)	Lage Zwaluwe (n = 20)	Bergen Op Zoom (n = 20)	Gennep (n = 12)	Schimmert (n = 3)	Total (n = 180)
*Leptospira* spp.	3 (2.4%, 0.5–6.9)	0 (0%, 0–16.8)	0 (0%, 0–16.8)	0 (0%, 0–26.5)	0 (0%, 0–70.8)	3 (1.7%, 0.4–4.8)
*Bartonella* spp.	42 (33.6%, 25.4–42.6)	3 (15%, 3.2–37.9)	16 (80%, 56.3–94.3)	6 (50%, 21.1–78.9)	2 (66.7%, 9.4–99.2)	69 (38.3%, 31.2–45.9)
*Hantaviridae*	63 (50.4%, 41.3–59.5)	10 (50%, 27.2–72.8)	9 (45%, 23.1–68.5)	6 (50%, 21.1–78.9)	1 (33.3%, 0.8–90.6)	89 (49.4%, 41.9–56.0)
Spotted fever group *Rickettsia*	0 (0%, 0–2.9)	1 (5%, 0.1–24.9)	0 (0%, 0–16.8)	0 (0%, 0–26.5)	0 (0%, 0–70.8)	1 (0.6%, 0.01–3.1)

******Anaplasma phagocytophilum*, *Babesia* spp., Neoehrlichia mikurensis, *Borrelia* spp., *Spiroplasma* spp. and *Francisella tularensis* were not detected.

## Data Availability

Supporting data can be found in Appendix A.

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
