# Peer review of "Occurrence of Rickettsia spp., Hantaviridae, Bartonella spp. and Leptospira spp. in European Moles (Talpa europaea) from the Netherlands"

_microorganisms, 2022, doi:10.3390/microorganisms11010041_

Round 1

Reviewer 1 Report

Line 25: “The European mole is a small” the Latin name of the animal should be used in this phrase: The European mole (Talpa europaea) is a small

Line 81: The font size for the name of the area on each figure should be increased!

Line 83: The Latin name of the animal should be used in the name of the drawing, as it will make the name of the drawing more informative.

Line 120: “MAFFT algorithm” – A reference should be provided.

Line 380: “38. FigTree: Tree figure drawing tool 1.4.4.” This "References" entry should be styled or removed.

Line 122-123: “in the FigTree v.1.4.4. program (Rambaut 2018).”  -> in the FigTree v.1.4.4. program (http://tree.bio.ed.ac.uk/software/figtree/) Accordingly, in line 380, remove "38. FigTree: Tree figure drawing tool 1.4.4."

Line 127: “95%CI 0.4-4.8” -> 95% CI 0.4-4.8

Line 131: “95%CI 31.2-45.9” -> 95% CI 31.2-45.9

Line 134: “95%CI 41.9-57.0” -> 95% CI 41.9-57.0),

Line 139: “95%CI 0.01-3.1” -> 95% CI 0.01-3.1

Line 144: 

“in moles from different” -> in moles Talpa europaea from different

”No. positive moles per capture location (Prevalence (%), 95% confidence interval)

95% confidence interval” -> No. positive moles per capture location (Prevalence (%), 95% confidence interval)

”(0%, 0-16.8))” -> (0%, 0-16.8)

“(0%, 0-70.8))” -> (0%, 0-70.8)

Line 145: “Anaplasma phagocytophilum, Babesia spp., Neoehrlichia mikurensis, Borellia spp., Spiroplasma” -> Anaplasma phagocytophilum, Babesia spp., Neoehrlichia mikurensis, Borellia spp., Spiroplasma

Line 146: “and Francisella tularensis were not” -> and Francisella tularensis were not

Line 155: Format Figure 2. Remove underscores and put Genbank accession numbers. Change the size of the fonts. In its current version, Figure 2 is visually perceived poorly, it needs to be corrected.

Line 170: Format Figure 2. Remove underscores and put Genbank accession numbers. Change the size of the fonts. In its current version, Figure 2 is visually perceived poorly, it needs to be corrected.

Line 197: “in 15/21 moles” – Please put into words what you want to convey by using "/" in this expression.

Author Response

Line 25: “The European mole is a small” the Latin name of the animal should be used in this phrase: The European mole (Talpa europaea) is a small

  • This is adapted in the new version.

Line 81: The font size for the name of the area on each figure should be increased!

  • The font size is increased in the new version. 

Line 83: The Latin name of the animal should be used in the name of the drawing, as it will make the name of the drawing more informative.

  • We didn’t find a good place to add it to the figure, but we added this to the Figure legend.

Line 120: “MAFFT algorithm” – A reference should be provided.

> This is added in the new version

Line 380: “38. FigTree: Tree figure drawing tool 1.4.4.” This "References" entry should be styled or removed.

  • This was adapted.

Line 122-123: “in the FigTree v.1.4.4. program (Rambaut 2018).”  -> in the FigTree v.1.4.4. program (http://tree.bio.ed.ac.uk/software/figtree/) Accordingly, in line 380, remove "38. FigTree: Tree figure drawing tool 1.4.4."

  • We adapted the reference, but kept this as a reference.

Line 127: “95%CI 0.4-4.8” -> 95% CI 0.4-4.8

  • This was adapted

Line 131: “95%CI 31.2-45.9” -> 95% CI 31.2-45.9

  • This was adapted

Line 134: “95%CI 41.9-57.0” -> 95% CI 41.9-57.0),

  • This was adapted

Line 139: “95%CI 0.01-3.1” -> 95% CI 0.01-3.1

  • This was adapted

Line 144: 

“in moles from different” -> in moles Talpa europaea from different

  • This was adapted.

”No. positive moles per capture location (Prevalence (%), 95% confidence interval)

95% confidence interval” -> No. positive moles per capture location (Prevalence (%), 95% confidence interval)

  • This was adapted

”(0%, 0-16.8))” -> (0%, 0-16.8)

  • This was adapted

“(0%, 0-70.8))” -> (0%, 0-70.8)

  • This was adapted

Line 145: “Anaplasma phagocytophilum, Babesia spp., Neoehrlichia mikurensis, Borellia spp., Spiroplasma” -> Anaplasma phagocytophilum, Babesia spp., Neoehrlichia mikurensis, Borellia spp., Spiroplasma

  • In the submitted version, these were already Italic. We will change it in the edited version. 

Line 146: “and Francisella tularensis were not” -> and Francisella tularensis were not

  • In the submitted version, these were already Italic. We will change it in the edited version.

Line 155: Format Figure 2. Remove underscores and put Genbank accession numbers. Change the size of the fonts. In its current version, Figure 2 is visually perceived poorly, it needs to be corrected.

> We have adapted this figure.

Line 170: Format Figure 2. Remove underscores and put Genbank accession numbers. Change the size of the fonts. In its current version, Figure 2 is visually perceived poorly, it needs to be corrected.

> We have adapted this figure.

Line 197: “in 15/21 moles” – Please put into words what you want to convey by using "/" in this expression.

  • This was adapted.

Reviewer 2 Report

The manuscript by Cuperus and collaborators investigates the occurrence of bacterial, viral and protozoal infections in moles from different areas of the Netherlands. Zoonotic disease ecology in insectivorous mammals deserves further research and moles are understudied compared to other mammals. Moreover, there is even more limited knowledge for this particular region. The study is affected by non-systematic sampling methods that prevent the generalization of the results to the source population. However, I recognize that it is very difficult to carry out a systematic sampling in these cases. Apart from that, results are supported by appropriate experimental approaches and the discussion and conclusion are consistent with the results. The manuscript is mostly well-written and easy to read. In my opinion the paper is worth to be published. However, some minor points should be addressed.

 General concept comments:

The authors declare "The aim of the current study was to assess the prevalence of a range of potentially zoonotic pathogens in a large sample of moles from the Netherlands". However, it is not clear whether the aim of the authors is really to assess the prevalence or rather (above all) that of evaluating the occurrence of those pathogens. It is important that the aim of the study adheres to what has actually been done.

A prevalence study generally requires a systematic approach to sampling. The authors do not give any indications regarding any planned pest control programs and/or mole sampling criteria. If this study is based on a convenience sample, no inference can be made about the source population of the sampled areas. Therefore, it must be clarified in the discussion that the estimated apparent prevalence values refer only to the sample of subjects analysed and not to the source population.

No adjustments for test performance of prevalence values were made. Therefore, reported prevalences are actually apparent prevalences. The authors should use the terms “apparent prevalence” instead of simply “prevalence” or this should at least be mentioned in the discussion.

 Specific comments:

Line 51: please change “have been” with “were” (you used the past tense in the rest of the manuscript)

Lines 130-131: please change “… an overall prevalence of 38.3% (Figure 1, Table 1, Table S2 95%CI 31.2-45.9)” with “… an overall prevalence of 38.3%, 95%CI 31.2-45.9 (Figure 1, Table 1, Table S2)”

Lines 135-137: please rephrase the following sentence to improve syntax “Hantavirus prevalence did not differ significantly between moles captured in 2018 around Urk (50.4%) or the southern capture locations from 2019 (47.3%)”. For example: “Hantavirus prevalence for moles captured in 2018 around Urk (50.4%) did not differ significantly from the mean prevalence value for those captured in 2019 from the southern locations (47.3%)”

Line 169: please change “showed respectively 82.6% and 83.1% similarity” with “showed 82.6% and 83.1% similarity, respectively”

Line 179: please change “suggesting a role” with “suggesting a potential role”

Line 186: delete “of these”

Line 189: please change “incidence of leptospirosis” with “incidence of human leptospirosis”

Lines 199-206: what you say in lines 199-202 “Sequences from the current study were closely related to these Bartonella sequences from Spain and Hungary. Possibly, these sequences could be part of a yet unidentified, mole-specific Bartonella species …” is in contrast with what you say further on in the text (lines 203-204 “… the sequences found are genetically quite similar to Bartonella spp. that have been shown to infect humans …”)

Line 232: please change “have” with “could have” and “small” with “minor”

Lines 232-234: it would be better to rephrase this period, the repetition and vagueness of "these pathogens" and of "a number of these pathogens" should be avoided

Line 238: please change “have been” with “were” (you used the past tense in the rest of the manuscript)

Lines 248-249: please rephrase “if so, this makes the European mole rare among land mammals” it could be understood that among land mammals moles are poorly represented

Line 380: please check the reference, which is indicated as Rambaut 2018 in the text.

Author Response

General concept comments:

... No adjustments for test performance of prevalence values were made. Therefore, reported prevalences are actually apparent prevalences. The authors should use the terms “apparent prevalence” instead of simply “prevalence” or this should at least be mentioned in the discussion.

  • This is a very true comment and we adapted the manuscript in several ways: changed “prevelalence” to “occurence” in the introduction; changed “prevalence”  to “apparent prevalence” in various parts of the manuscript, and we discuss this limitation in the discussion. We have not changed “prevalence” in all sentences in the results to keep the text easy to read, but together with the discussion, we think it should be clear to the reader.

 Specific comments:

Line 51: please change “have been” with “were” (you used the past tense in the rest of the manuscript)

  • This was adapted.

Lines 130-131: please change “… an overall prevalence of 38.3% (Figure 1, Table 1, Table S2 95%CI 31.2-45.9)” with “… an overall prevalence of 38.3%, 95%CI 31.2-45.9 (Figure 1, Table 1, Table S2)”

  • This was adapted.

Lines 135-137: please rephrase the following sentence to improve syntax “Hantavirus prevalence did not differ significantly between moles captured in 2018 around Urk (50.4%) or the southern capture locations from 2019 (47.3%)”. For example: “Hantavirus prevalence for moles captured in 2018 around Urk (50.4%) did not differ significantly from the mean prevalence value for those captured in 2019 from the southern locations (47.3%)”

  • This was adapted.

Line 169: please change “showed respectively 82.6% and 83.1% similarity” with “showed 82.6% and 83.1% similarity, respectively”

  • This was adapted.

Line 179: please change “suggesting a role” with “suggesting a potential role”

  • This was adapted.

Line 186: delete “of these”

  • This was adapted.

Line 189: please change “incidence of leptospirosis” with “incidence of human leptospirosis”

  • This was adapted.

Lines 199-206: what you say in lines 199-202 “Sequences from the current study were closely related to these Bartonella sequences from Spain and Hungary. Possibly, these sequences could be part of a yet unidentified, mole-specific Bartonella species …” is in contrast with what you say further on in the text (lines 203-204 “… the sequences found are genetically quite similar to Bartonella spp. that have been shown to infect humans …”)

  • We adapted mole-specific to mole-associated

Line 232: please change “have” with “could have” and “small” with “minor”

  • This was

Lines 232-234: it would be better to rephrase this period, the repetition and vagueness of "these pathogens" and of "a number of these pathogens" should be avoided

  • The paragraph was adapted and a list of pathogens tested in the studies referenced was added.

Line 238: please change “have been” with “were” (you used the past tense in the rest of the manuscript)

  • This was adapted.

Lines 248-249: please rephrase “if so, this makes the European mole rare among land mammals” it could be understood that among land mammals moles are poorly represented

  • This was changed to “ distinctive” .

Line 380: please check the reference, which is indicated as Rambaut 2018 in the text.

  • This was adapted.

Reviewer 3 Report

Comments for authors:

Title:

Kindly, modify it to be short.

Abstract:

·        L10: there is little knowledge……….  there is a little knowledge.

·        L13: …We also found, infections with Leptospira spp. (1.7%)…Please make all writing methods in passive form like Leptospira spp. infection was found in …….  Kindly, take in consideration across the whole manuscript.

·        L13- L15: Please include the number of positive cases out of the total tested animals adding to the percentage of infection like……../….. (….%).

Introduction:

·        L31: ….to carry a diversity of zoonotic pathogens…….. Kindly add examples of these zoonotic pathogens.

·        L32: (Krijger et al. 2020; Szekeres et al. 2019; Špitalská et al. 2017). Please follow the journal style regarding the references arrangement across the whole manuscript. Also, in L39: (Špitalská et al. 2017; Szekeres et al. 2019; Gil et al. 2010) which arrangement style is ok???

Material and methods:

·        L70 -72:  add the latitude and longitude of the mentioned provinces.

·        L77: …….After capture, moles were stored at -20°C until dissection. I am interested to know wither the capture process resulted in killing of the captured animals, or it was alive and then it  should be euthanized ??? please clarify??

·        L99: ……..3 μl of sample……..  did you mean the DNA ????  , please add it

Results:

·        L152: (100% similarity, sequence not shown in Figure 2 because of short fragment length), Why did not you use another pair of primers to have long amplicon??

·        Based on Table S2 in supplementary file, There was a mixed infection of Bartonella spp. and Hantaviridae in the same animal. Please refer to this and its percentage in the result section and also in Discussion part.

Discussion:

·        L250-252: Based on this paragraph, it was better to check presence of any tick types on the trapped animals and detect these pathogens in the extracted DNA???? This would improve the potency of this manuscript.

 Figure (1):

·        Please add the key for the basic directions (East, West, North, and South).

·        Did you use specific software for creating maps or figure?  Please mention it.

References:

·        the journal style regarding the references arrangement should be followed across the whole manuscript.

Author Response

Title:

Kindly, modify it to be short.

  • We have tried several options, but we think the current title is the most informative, while still being of a reasonable length. If we leave out pathogens, host or location, we think the title doesn’t represent the information in the article well, so we would like to keep the title as is.

Abstract:

L10: there is little knowledge……….  there is a little knowledge.

  • We adapted this to “ little is known”

L13: …We also found, infections with Leptospira spp. (1.7%)…Please make all writing methods in passive form like Leptospira spp. infection was found in …….  Kindly, take in consideration across the whole manuscript.

  • We have checked this with the editor, but microorganisms has no preference regarding the passive or active voice. To meet the reviewer’s preference, we have changed several sentences throughout the manuscript from the active to the passive voice, but in some cases, e.g. the material and methods and the conclusions, we thought the active voice was more clear.

L13- L15: Please include the number of positive cases out of the total tested animals adding to the percentage of infection like……../….. (….%).

  • We added the absolute number of infected moles per pathogen. Since they were all tested in the 180 moles that are mentioned in the prior sentence, we didn’t include the total.

Introduction:

L31: ….to carry a diversity of zoonotic pathogens…….. Kindly add examples of these zoonotic pathogens.

  • We included two examples.

L32: (Krijger et al. 2020; Szekeres et al. 2019; Špitalská et al. 2017). Please follow the journal style regarding the references arrangement across the whole manuscript. Also, in L39: (Špitalská et al. 2017; Szekeres et al. 2019; Gil et al. 2010) which arrangement style is ok???

  • We adapted the reference style to MDPI Chicago style.

Material and methods:

L70 -72:  add the latitude and longitude of the mentioned provinces.

  • We deleted the mention of the provinces, as these were not used later on in the manuscript. We now only mention the region of the country and the location. We do not have the precise coordinates of all the study locations, but together with Figure 1, we think it is clear for readers where the various locations are in the Netherlands.

L77: …….After capture, moles were stored at -20°C until dissection. I am interested to know wither the capture process resulted in killing of the captured animals, or it was alive and then it  should be euthanized ??? please clarify??

  • It was lethal trapping. This was added to the first sentence of the paragraph.

L99: ……..3 μl of sample……..  did you mean the DNA ????  , please add it

  • Yes. This was added.

Results:

L152: (100% similarity, sequence not shown in Figure 2 because of short fragment length), Why did not you use another pair of primers to have long amplicon??

  • We have used the same primer pair that we have used in other studies. However, the overlap with the sequence from this Spanish samples was very short.

Based on Table S2 in supplementary file, There was a mixed infection of Bartonella spp. andHantaviridae in the same animal. Please refer to this and its percentage in the result section and also in Discussion part.

  • The information about the mixed infections in the moles was added to the Results sections (L154-156).

Discussion:

L250-252: Based on this paragraph, it was better to check presence of any tick types on the trapped animals and detect these pathogens in the extracted DNA???? This would improve the potency of this manuscript.

  • Though we did not perform a thorough look for each mole, we cannot recall that we saw any ticks on the moles during dissections of the moles. Because we received carcasses from the mole trappers, it is very likely any potential ticks would have left the carcass already. Therefor, it was not possible to check the ticks themselves for pathogens.

 Figure (1):

Please add the key for the basic directions (East, West, North, and South).

> This is added to the new figure. 

Did you use specific software for creating maps or figure?  Please mention it.

  • We used R. This was added.

References:

the journal style regarding the references arrangement should be followed across the whole manuscript.

  • This was adapted.